# A network of human functional gene interactions from knockout fitness screens in cancer cells

Eiru Kim[1], Merve Dede[1,2], Walter F Lenoir[1,2] , Gang Wang[1], Sanjana Srinivasan[1,2], Medina Colic[1,2], Traver Hart[1,3]

**Genetic interactions mediate the emergence of phenotype from genotype. The systematic survey of genetic interactions in yeast showed that genes operating in the same biological process have highly correlated genetic interaction profiles, and this observation has been exploited to infer gene function in model organisms. Such assays of digenic perturbations in human cells are also highly informative, but are not scalable, even with CRISPR-mediated methods. As an alternative, we developed an indirect method of deriving functional interactions. We show that genes having correlated knockout fitness profiles across diverse, non-isogenic cell lines are analogous to genes having correlated genetic interaction profiles across isogenic query strains and similarly imply shared biological function. We constructed a network of genes with correlated fitness profiles across 276 high-quality CRISPR knockout screens in cancer cell lines into a "coessentiality network," with up to 500-fold enrichment for co-functional gene pairs, enabling strong inference of gene function and highlighting the modular organization of the cell.**

## Introduction

Genetic interactions govern the translation of genotype to phenotype at every level, from the function of subcellular molecular machines to the emergence of complex organismal traits. In the budding yeast *Saccharomyces cerevisiae*, systematic genetic deletion studies showed that only ~1,100 of its ~6,000 genes (~20%) were required for growth under laboratory conditions (Giaever et al, 2002). A systematic survey of digenic knockouts, however, yielded hundreds of thousands of gene pairs whose double knockout induced a fitness phenotype significantly more severe (synergistic genetic interactions) or less severe (suppressor interactions) than expected from each gene's single mutant fitness (Tong et al, 2001; Costanzo et al, 2010, 2016), with triple-mutant screens adding yet another layer of complexity (Kuzmin et al, 2018). When trying to decipher the genetic contribution to as simple a phenotype as fitness, then, there are vastly more candidate explanations involving genetic interactions than monogenic fitness effects. Moreover, the impact of each gene variant not only depends on the sum of all other genetic variants in the cell but also is strongly influenced by the cell's environment (Hillenmeyer et al, 2008; Bandyopadhyay et al, 2010).

Patterns of genetic interaction are deeply informative. Genetic interactions frequently occur either within members of the same pathway or process ("within pathway interactions") or between members of parallel pathways ("between pathway interactions") (Kelley & Ideker, 2005). When assayed systematically, the result is that genes that operate in the same biological process tend to interact genetically with the same sets of other genes in discrete, related pathways, culminating in highly correlated genetic interaction profiles across a diverse panel of genetic backgrounds or "query strains." This observation has been exploited extensively to infer gene function in model organisms and, on a smaller scale, in human cells based on similarity of genetic interaction profiles (Lehner et al, 2006; Horn et al, 2011; Bassik et al, 2013; Kampmann et al, 2013; Roguev et al, 2013; Costanzo et al, 2016). Therefore, beyond the specific interactions themselves, a gene's pattern of fitness phenotypes across a diverse set of backgrounds can inform our knowledge of that gene's function.

Translating these concepts into human cells has proved biologically and technically challenging. The *S. cerevisiae* genome has less than one-third the number of protein-coding genes as humans, and despite the quantum leap in technology that the CRISPR/Cas system offers to mammalian forward genetics, yeast remains far simpler to perturb reliably in the laboratory. Several groups have applied digenic perturbation technologies, using both shRNA and CRISPR, to find cancer genotype-specific synthetic lethals for drug targeting (Wong et al, 2016; Du et al, 2017; Han et al, 2017; le Sage et al, 2017; Shen et al, 2017; Najm et al, 2018) and to identify genetic interactions that enhance or suppress phenotypes related to drug and toxin resistance (Bassik et al, 2013; Roguev et al, 2013; Jost et al, 2017). The current state of the art in CRISPR-mediated gene perturbation relies on observations from three independent gRNA targeting each gene, or nine pairwise perturbations for each gene pair, plus non-targeting or other negative controls. The largest

---

[1]Department of Bioinformatics and Computational Biology, The University of Texas MD Anderson Cancer Center, Houston, TX, USA   [2]UTHealth Graduate School of Biomedical Sciences, The University of Texas MD Anderson Cancer Center, Houston, TX, USA   [3]Department of Cancer Biology, The University of Texas MD Anderson Cancer Center, Houston, TX, USA

Correspondence: traver@hart-lab.org

such mapping to date puts the scale of the problem in stark terms: Han et al (2017) use a library of 490,000 gRNA doublets—seven times larger than a latest generation whole-genome, single-gene knockout library—to query all pairs of 207 target genes or ~0.01% of all gene pairs in the human genome (Han et al, 2017).

An additional dimension of the scale problem is that of backgrounds. Whereas one strain of yeast was systematically assayed in fixed media and environmental conditions to create a reference genetic interaction network, no such reference cell exists for humans. Indeed first-generation whole-genome CRISPR screens in cancer cell lines demonstrated that one of the features associated with the hugely increased sensitivity of CRISPR over shRNA (Hart et al, 2014, 2015) was the ability to resolve tissue- and genetic-driven differences in gene essentiality and the unexpected variation in gene essentiality in cell lines with ostensibly similar genetic backgrounds (Wang et al, 2014; Hart et al, 2015).

Nevertheless, small-scale, targeted genetic interaction screens in human cells using both shRNA and CRISPR showed that the architecture of the genetic interaction network holds true across species. Positive and negative genetic interactions within pathways and between related biological processes yield a correlation network with the same properties: genes with similar profiles of genetic interaction across different backgrounds are often in the same process or complex, providing a strong basis for inference of gene function (Horn et al, 2011; Bassik et al, 2013, 2013; Kampmann et al, 2013, 2014; Roguev et al, 2013). Because digenic perturbation screens are difficult to scale, we considered whether indirect methods of determining functional genomic information might be effective on a large scale. Since then, whole-genome CRISPR knockout screens have been performed in more than 400 cancer and immortalized cell lines, with the bulk coming from the Cancer Dependency Map project using standardized protocols and reagents (Aguirre et al, 2016; Meyers et al, 2017; Tsherniak et al, 2017). We hypothesized that genes having correlated knockout fitness profiles across diverse cell lines would be analogous genes having correlated genetic interaction profiles across specified query backgrounds in the same cells, and would similarly imply shared biological function. This extends a concept explored by Wang et al (2017), at a small scale, and more deeply by Pan et al (2018) to discover protein complexes from correlated fitness profiles. We constructed a network of genes with correlated essentiality scores into a "coessentiality network," from which we identified clusters of genes with high functional coherence. The network provides powerful insight into functional genomics, cancer targeting, and the capabilities and limitations of CRISPR-mediated genetic screening in human cell lines.

## Results and Discussion

We considered CRISPR and shRNA whole-genome screen data from multiple libraries and laboratories: Avana (Doench et al, 2014; Meyers et al, 2017), GeCKOv2 (Aguirre et al, 2016), TKO (Hart et al, 2015, 2017a; Steinhart et al, 2017), Sabatini (Wang et al, 2014, 2017), the Moffat shRNA library (Koh et al, 2012; Marcotte et al, 2012, 2016; Medrano et al, 2017), and other large data sets (McDonald et al, 2017;

Tsherniak et al, 2017) (Fig 1A and Table S1). From raw read count data, we used the BAGEL pipeline (described in Hart & Moffat (2016) and improved here; see the Materials and Methods section) to generate Bayes factors for each gene in each cell line. We removed non-targeting and nonhuman gene controls and quantile-normalized each data set to mitigate screen quality bias, yielding an essentiality score where a positive value indicates a strong knockout fitness defect, and a negative value generally implies no phenotype (see the Materials and Methods section for details). Each gene, therefore, has an "essentiality profile" of its scores across the screens in that data set.

For each data set, we ranked gene pairs by correlated essentiality profiles and measured the enrichment for co-functional pairs (see the Materials and Methods section). We used the log-likelihood score (LLS) to describe the significance of enrichment. Gene pairs are ranked by Pearson correlation, grouped into bins of 1,000 pairs, and each bin is evaluated for the relative abundance of genes annotated to be in the same KEGG pathway ("true positives") versus genes annotated to be in different pathways ("false positives"). Data from Meyers et al (2017), where CRISPR knockout screens were conducted using the Avana library in 342 cancer cell lines, showed the strongest enrichment for co-functional gene pairs (Fig 1B), likely because of the relatively high quality of the screens (Fig S1) as well as the lineage and genetic diversity of the cells being screened. In contrast, screens from Wang et al (2014), (2017) were equally of high quality but were performed only in 17 acute myeloid leukemia (AML) cell lines with correspondingly limited diversity. To further increase the co-functionality signal, we removed screens with poor performance and only considered genes that were hits in at least three of the remaining screens; filtering resulted in an additional twofold enrichment for co-functional gene pairs (Figs 1B and S1). The filtered data from Meyers et al (2017) (n = 276 cell lines; 5,387 genes; hereafter "Avana data") was used for all subsequent analysis. We selected gene pairs with a Bonferroni-corrected $P$-value < 0.05 and combined them into a network, the Cancer Coessentiality Network, containing 3,327 genes connected by 68,813 edges (Fig 1C). It has been observed that off-target effects of promiscuous gRNA can influence essentiality scores (Fortin et al, 2018). We evaluated interactions whether there is a correlation drop after removing all sgRNAs with 1-bp mismatch against interactors. We marked all correlation drops as off-target–suspected interactions (Table S5). The resulting network is highly modular, with clusters showing strong functional coherence, similar to the networks directly inferred from correlated yeast genetic interaction profiles (Costanzo et al, 2010, 2016).

### Essential genes specific to oncogenic contexts

The data underlying the Cancer Coessentiality Network is derived from well-characterized cancer cell lines from 30+ lineages, representing the major oncogenic mutation profiles common to those cancers. We expected that many clusters in the network could, therefore, be associated with specific tissues and cancer-relevant genotypes. By testing cluster-level essentiality profiles for tissue specificity (see the Materials and Methods section), we identified only a small number of clusters that correspond to tissue-specific cancers (Fig 2A), which in turn contain the characteristic oncogenes. For example, cluster 14 (Fig 2B) consists of *BRAF* and related genes

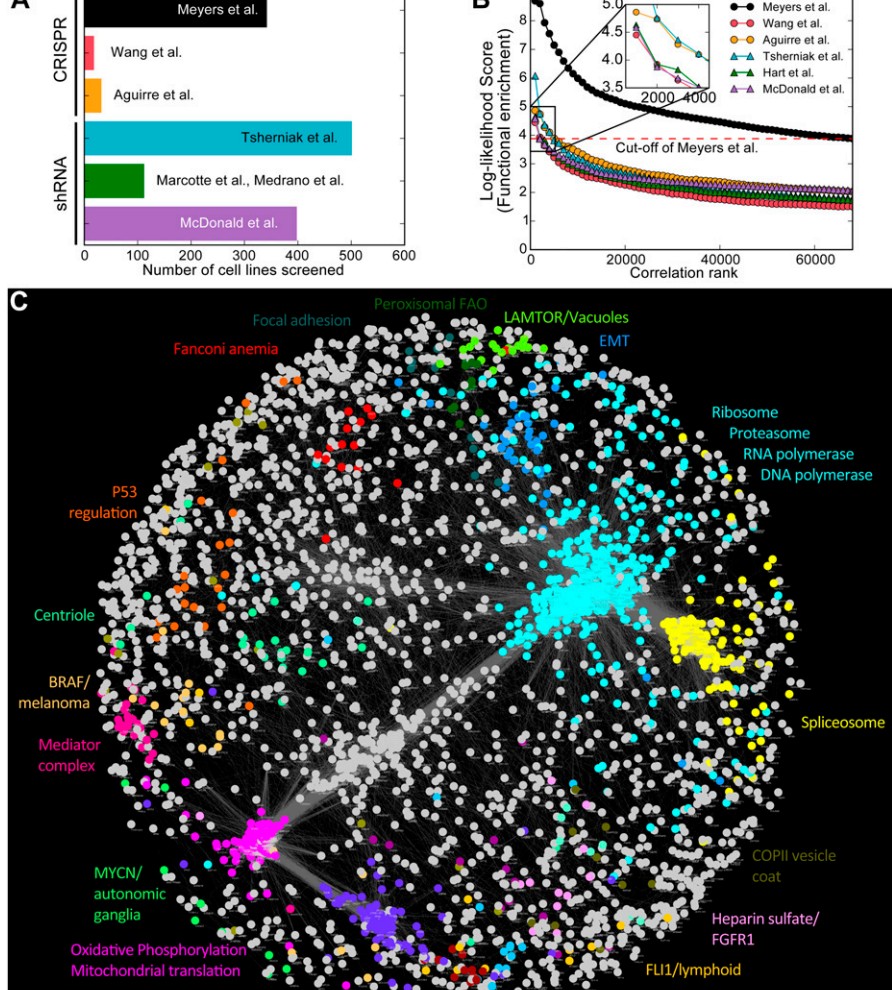

**Figure 1. The coessentiality network.**
**(A)** CRISPR and shRNA screens analyzed for this study. **(B)** Measuring functional enrichment. For each data set, pairwise correlations of knockout/knockdown fitness profiles were ranked, binned (n = 1,000), and measured for enrichment for shared KEGG terms. Data from Meyers et al (2017) ("Avana data") carry significantly more functional information than other data sets. **(C)** The Cancer Coessentiality Network, derived from Avana data, contains 3,483 genes connected by 68,813 edges. Selected modules, derived by an unbiased clustering algorithm and color-coded, demonstrate the functional coherence of the network.

that are highly specific to *BRAF*-mutated melanoma cells ($P < 10^{-12}$; Fig 2C). The cluster contains other elements of the mitogen-activated protein kinase (MAPK) pathway (*MAP2K1*, *MAPK1*, and *DUSP4*), indicating their essentiality in *BRAF*-mutant cells, supporting efforts to incorporate ERK inhibitors into combinatorial therapies to overcome resistance to targeted *BRAF* treatments (Smalley & Smalley, 2018). This example highlights the utility of this indirect approach to identify synthetic lethal interactions: genes co-essential with oncogenes are synthetic lethals. Beyond the downstream elements of the MAPK pathway itself, the *BRAF*/melanoma cluster also contains the transcription factors (TFs), melanogenesis-associated transcription factor, the developmental Sry-related box (Sox) gene *SOX10*, and mesenchymal marker *ZEB2*, indicative of the non-epithelial origin of melanocyte cells and providing insight into the genetic requirements for tissue differentiation in this lineage.

Similar observations hold for other tissue-specific oncogenes. Cluster 17, essential in lymphoid cell lines ($P < 10^{-7}$), contains oncogene *FLI1* and tissue-specific TF *MYB* (Fig 2D and E), and cluster 38 is enriched for ovarian cancer cells ($P < 10^{-7}$) and carries lineage-

specific TF *PAX8*, previously shown to be essential in these cells (Cheung et al, 2011) (Fig 2F and G). Cluster 75, essential in colorectal cancer cells ($P < 10^{-9}$), contains β-catenin (*CTNNB1*) and TF partner *TCF7L2* (Fig 2H and I); both are linked to E2 ubiquitin ligase *UBE2Z*, which mediates *UBA6*-specific suppression of epithelial-to-mesenchymal transition (EMT) (Liu et al, 2017), indicating a functional linkage with β-catenin signaling. Additional tissue- and oncogene-driven clusters delineating oncogenic receptor tyrosine kinase (RTK) and MAP kinase signaling, joint cyclin/cyclin-dependent kinase dependencies, and a set of genes enriched in nuclear lamina maintenance that are preferentially essential in glioblastoma cells are shown in Fig S2.

Neuroblastoma cells require *MYCN*, the neuroblastoma-specific paralog of the *MYC* oncogene (Huang & Weiss, 2013), as well as nervous system developmental TF *SOX11* (Potzner et al, 2010) (Fig 2J). Interestingly, *MYC* is highly essential in virtually all non-neuroblastoma cell lines, resulting in a relatively uncommon anti-correlation in *MYC* and *MYCN* essentiality profiles (r = −0.49; $P < 10^{-17}$; Fig 2K). While this negative correlation is driven by mutual exclusivity in tissues, we also observe anti-correlation between

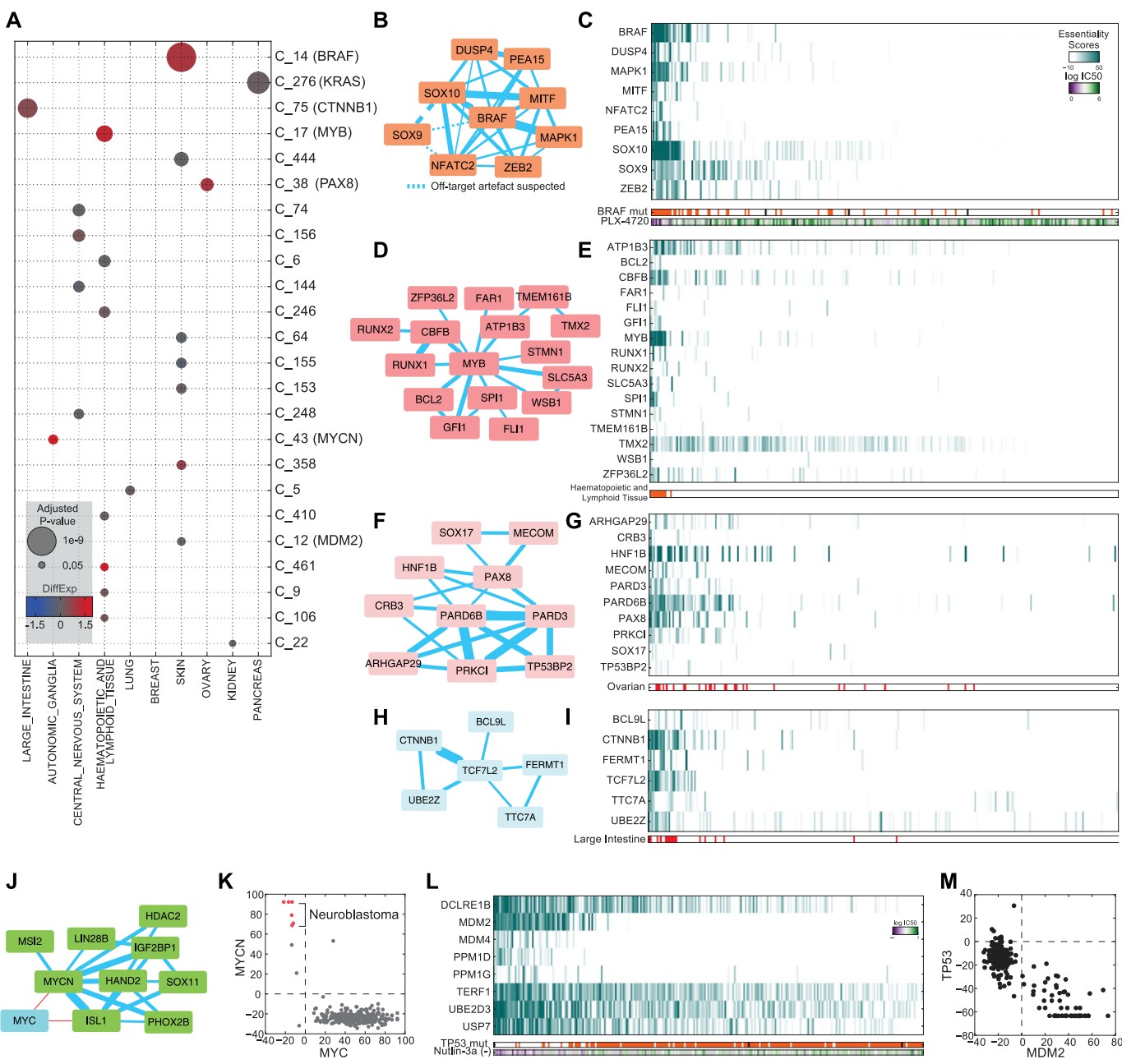

**Figure 2. Cancer-specific features of the network.**
**(A)** Clusters of genes were evaluated for tissue specificity (size of circles) and differential mRNA expression of genes in the cluster (color of circles). **(B)** Cluster 14 (BRAF cluster); nodes are genes in cluster and edges reflect the strength of correlation of fitness profile. Dashed lines indicate suspected off-target interactions (see the Materials and Methods section) **(C)** Heat map of essentiality profiles of genes in BRAF cluster, ranked by median essentiality score. Gene essentiality in the cluster is associated with PBRAF mutation ($P < 10^{-23}$) and sensitivity to BRAF inhibitor PLX-4720 ($P < 10^{-7}$). **(D, E)** Network and heat map of MYB-related cluster. **(F, G)** PAX8-associated cluster. **(H, I)** B-catenin cluster. **(J)** MYCN neuroblastoma cluster is anti-correlated with MYC. **(K)** MYC and MYCN essentiality is mutually exclusive. **(L)** MDM2 cluster heat map is associated with TP53 mutation status ($P < 10^{-13}$) and sensitivity to Nutlin-3a ($P < 10^{-14}$). **(M)** MDM2 versus TP53 essentiality. TP53 essentiality scores < −50 indicate tumor suppressor role.

tumor suppressors and their repressors in the same cells. CRISPR knockout of tumor suppressors in cells carrying wild-type alleles frequently results in increased growth rate, which manifests as extreme negative essentiality scores. Melanoma cells with wild-type *TP53* show these extreme negative values, resulting in strong anti-correlation with *TP53* suppressors *MDM2* (r = −0.86, $P < 10^{-81}$), *MDM4* (r = −0.61, $P < 10^{-28}$), and *PPM1D* (r = −0.72, $P < 10^{-44}$) (Fig 2L and M).

Although *TP53* shows the characteristic extreme negative essentiality score of a tumor suppressor gene in wild-type backgrounds, surprisingly, it causes a growth defect when knocked out in three cell lines: HCT1143 breast cancer, PC14 lung cancer, and NB4 AML cells (Fig S3A). All three carry the R248Q oncogenic mutation; in fact, R248Q is weakly predictive of *TP53* essentiality generally, and strongly predictive when it is the only P53 mutation detected

(Fig S3B). Nor is this the only case where a tumor suppressor in one background is an essential gene in another: the von Hippel–Landau tumor suppressor gene *VHL* shows no phenotype in renal cancer cells, where the gene is nearly universally deleted, but is essential specifically in BTFC-909 renal carcinoma cells which lack the characteristic Chr3 copy loss (Sinha et al, 2017). In contrast, *VHL* shows a fitness defect when knockout out in most other backgrounds (Fig S3C). The essentiality profile for *VHL* is strongly correlated with *EGLN1* (commonly called *PHD2*), an oxygen sensor that hydroxylates hypoxia response genes *HIF1A* and *HIF2A*, marking them for degradation by the *VHL* complex in normoxic environments (Berra et al, 2003). *EGLN1* essentiality is overrepresented in melanoma cells ($P < 10^{-4}$, rank-sum test; essential in 14 of 22 skin cancer cell lines).

### A high-precision functional interaction map of human genes

These examples indicate the breadth and precision of the coessentiality network but represent results from hypothesis-guided queries. In an effort to learn novel associations from the data, we

tested each cluster for its correlation with cell lineage as well as correlation with gene expression, mutation, and copy number amplification of all genes both inside and outside the cluster to identify underlying molecular genetic drivers of modular, emergent essentiality. We identified 270 genes in 30 clusters whose essentiality profiles strongly correlated with their own copy number profiles but not their expression profiles (Fig 3A). As copy number amplification is a known source of false positives in CRISPR screens, we labeled these clusters as amplification artifacts. An additional 56 genes in 11 clusters showed significant association with both copy number and expression. These clusters notably include *KRAS* amplifications in pancreatic and colorectal cancer (cluster 276), *ERBB2* amplifications in breast and other cancers (cluster 52), and *CCNE1* overexpression/*RB1* mutation (cluster 101), consistent with well-studied patterns of oncogenesis. All network cluster annotations can be found in the master annotation file (Table S7).

Given the underlying data, it is not surprising that oncogenic signatures are clearly evident in the coessentiality network. However, the vast majority of the network structure does not appear to be driven by

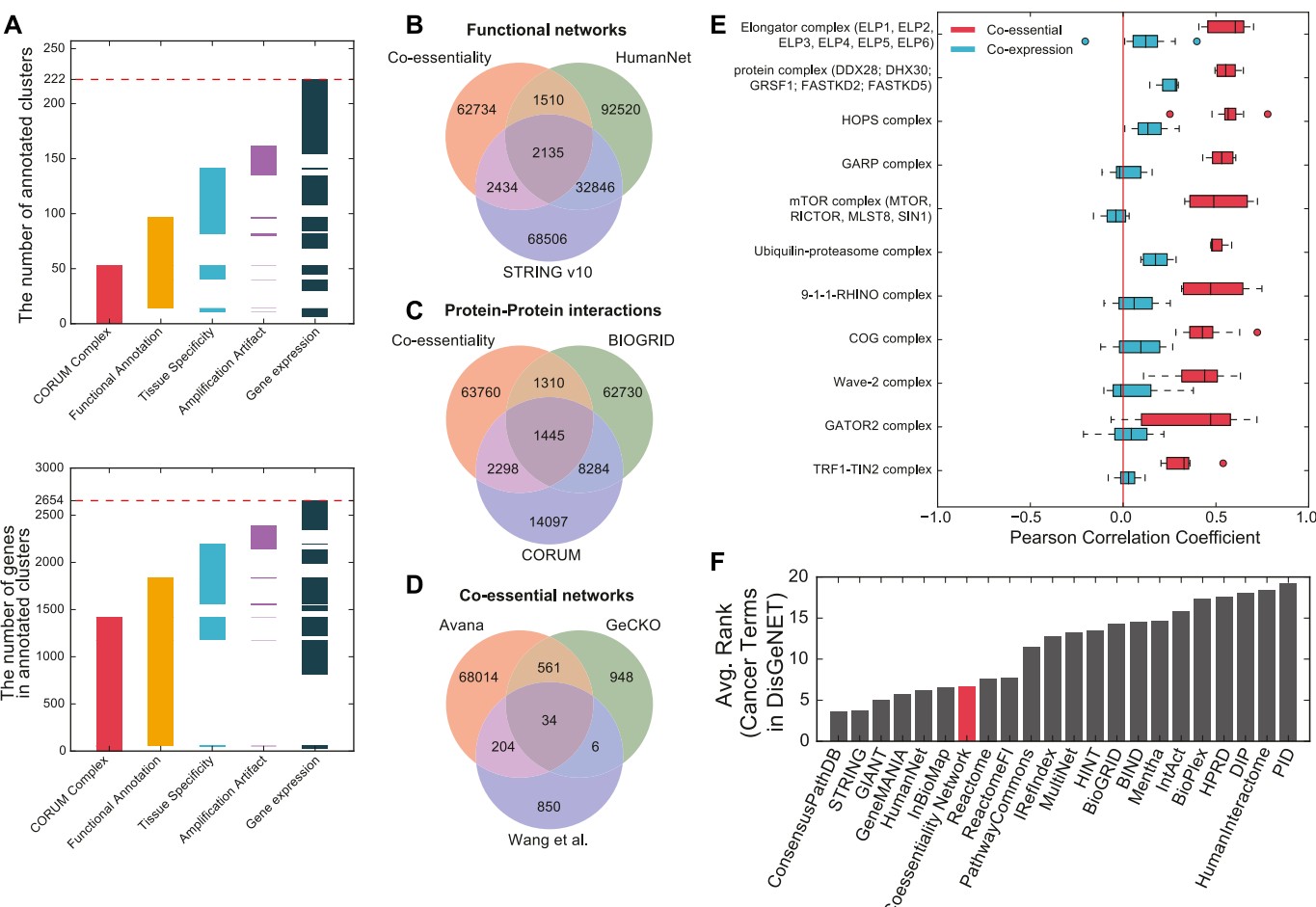

**Figure 3. Beyond cancer: characterization of the coessentiality network.**
**(A)** Number of clusters (top) and total genes in clusters (bottom) showing strong association with annotated protein complexes, biological function, tissue specificity, amplification-induced CRISPR artifacts, and differential expression of genes. **(B–D)** Comparing the Avana coessentiality network with other functional (B), protein–protein (C), and coessentiality networks (D) shows the unique information contained in our network. Each number indicates the number of interactions. **(E)** For some protein complexes, coessentiality is a better predictor of co-complex membership than co-expression. **(F)** The coessentiality network is a powerful predictor of cancer pathways (Huang et al, 2018) compared with other databases and networks (lower rank is better).

tissue specificity or mutational signatures. The network contains information complementary to prior functional (Fig 3B) and physical (Fig 3C) interaction networks, and the network derived from Avana data exhibits far greater coverage than equivalent networks from the GeCKOv2 subset of Project Achilles (Aguirre et al, 2016) or Wang (Wang et al, 2017) AML-specific data (Fig 3D). Nevertheless, the remaining network modules show strong functional coherence (Fig 3A). We also compared our network with previously published analyses of the same CRISPR screen data inferring protein complexes (Pan et al, 2018) and genetic interactions (Rauscher et al, 2017). The coessentiality network substantially expands on Pan et al (2018) and is largely orthogonal to Rauscher et al (2017), owing to the fundamentally different approaches used to generate these networks (Fig S4).

Coessentiality often proves a stronger predictor of complex membership than coexpression (Fig 3E), and this signature is reflected in the network clusters we identified. Indeed, 53 clusters, comprising 1,422 genes, show enrichment for CORUM-annotated protein complexes at $P$-value < $10^{-6}$, and fitness profiles have been used to implicate additional members of protein complexes (Pan et al, 2018). However, this holds only for genes whose knockout fitness defects vary across cell lines; coessentiality of core essential genes is poorly predictive of co-complex membership (Fig S5). All 53 CORUM-annotated clusters, plus an additional 44 clusters containing 413 genes (totaling 97 network modules with 1,835 genes), show enrichment for GO biological process, cellular component, KEGG pathway, or Reactome pathway annotations at a similarly strict threshold. In addition, we evaluated the relative performance of the coessentiality network by measuring its ability to recover cancer gene sets using DisGeNET (Huang et al, 2018). The coessentiality network ranks comparably with other large functional networks (Fig 3F), although starting from a much smaller data set, suggesting that the coessentiality network explains not only protein complexes but also cancer pathways, including interactions between protein complexes and signaling transduction.

Epistatic interactions frequently underlie covariation in fitness profiles (Phillips, 2008). Cluster 2 is highly enriched for genes involved in the mitochondrial electron transport chain, including 30 of 48 genes encoding subunits of NADH dehydrogenase complex (ETC Complex I; $P$ < $10^{-42}$) plus additional subunits of all other ETC complexes. The cluster also contains 49 of 51 subunits of the mitochondrial large ribosomal subunit ($P$ < $10^{-87}$), 23 of 25 members of the small subunit ($P$ < $10^{-39}$), plus 20 mitochondrion-specific tRNA synthases ($P$ < $10^{-20}$). This mitochondrial translation machinery is required for the synthesis of proteins in the ETC complexes. These genes' inclusion in this cluster, where their essentiality profiles are correlated with those of the complexes they support, reflects a fundamental feature of saturating genetic screens: the essentiality of a given enzyme or biological process is matched by the essentiality of the cellular components required for the biogenesis and maintenance of that process.

We observe numerous additional instances of such epistatic interactions that highlight functional relationships. For example, glutathione peroxidase gene *GPX4* shows highly variable essentiality across cell lines (Fig 4A and C). *GPX4* is a selenoprotein that contains the cysteine analog selenocysteine (Sec), the "21st amino acid," at its active site. Coessential with *GPX4* are all the genes required for conversion of serine-conjugated tRNA[Ser] to selenocysteine-conjugated tRNA[Sec] (*PSTK, SEPHS2,* and *SEPSECS*), as well as selenocysteine-specific elongation factor *EEFSEC*, which guides Sec-tRNA[Sec] to specific UGA codons (Fig 4b) (Schoenmakers et al, 2016). Cellular dependence on *GPX4* was recently shown to be associated with mesenchymal state (Viswanathan et al, 2017), and our analysis corroborates this observation: we find that *GPX4* essentiality is higher in cells expressing mesenchymal marker *ZEB1* ($P$ < $10^{-5}$; Fig 4D). However, *GPX4* sensitivity is more strongly associated with low expression of *GPX2*, another member of the glutathione peroxidase family (Fig 4E), suggesting a candidate synthetic lethal interaction between *GPX2* and *GPX4*.

Similarly, a pair of genes, *ACOX1* and *HSD17B4*, which encode three of the four enzymatic steps in peroxisomal fatty acid B-oxidation

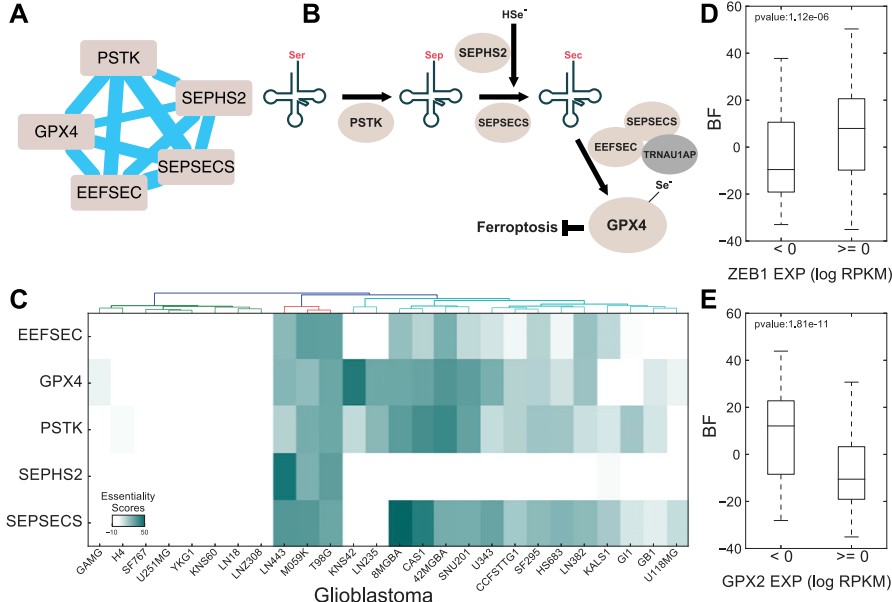

**Figure 4.  GPX4 cluster.**
**(A, B)** Glutathione peroxidase GPX4, a selenoprotein, is strongly clustered with genes involved in the selenocysteine conversion pathway (B). **(C)** Entire GPX4 cluster shows marked differential essentiality in glioblastoma cell lines. **(D, E)** Cellular requirement for GPX4 is associated with ZEB1 expression, as previously reported, but (E) GPX2 expression is more strongly predictive.

(FAO), are found in a cluster with 10 PEX genes involved in peroxisome biogenesis, maintenance, and membrane transport (Fig 5A and B). The cluster shows a discrete pattern of essentiality, preferentially in lung cells (essential in 6/42 lung cancer lines in the Avana data; Fig 5C) but also appearing intermittently in other lineages. Notably, this cluster is intact in the network generated from Aguirre et al (2016) (Fig 5D), although it arises in pancreatic cells rather than lung cells. The small number of cell lines showing the PEX phenotype preclude a robust identification of predictive biomarkers; neither the Avana/lung cluster nor the GeCKO/pancreatic cluster is significantly associated with mutational or lineage-specific features, and differential gene expression analysis yielded no functionally coherent results.

## A network of interactions between biological processes

Although individual clusters show high functional coherence, the network of connections between clusters offers a unique window into process-level interactions in human cells. The peroxisomal FAO cluster is strongly connected to another functionally coherent module containing 12 genes, 10 of which are tightly connected to other members of the cluster (Fig 5A). Those 10 include seven genes whose proteins reside in the ER, five of which regulate cholesterol biosynthesis via posttranslational modification of sterol regulator element-binding proteins (SREBPs). The remaining three genes, *DHRS7B*, *TMEM41A*, and *C12orf49*, are largely or completely uncharacterized; their strong association with other genes in this cluster implicates a role in the SREBP maturation pathway. Both the peroxisomal FAO cluster and the SREBP maturation cluster are linked with a module containing *RAB18*, a RAS-related GTPase involved in Golgi-to-ER retrograde transport, as well as its associated GTPase-activating proteins (GAP), *RAB3GAP1* and *RAB3GAP2*, and guanine nucleotide exchange factor (GEF), *TBC1D120* (Feldmann et al, 2017).

A similar network of modules describes the regulation of the mechanistic target of rapamycin (mTOR), in particular, its detection

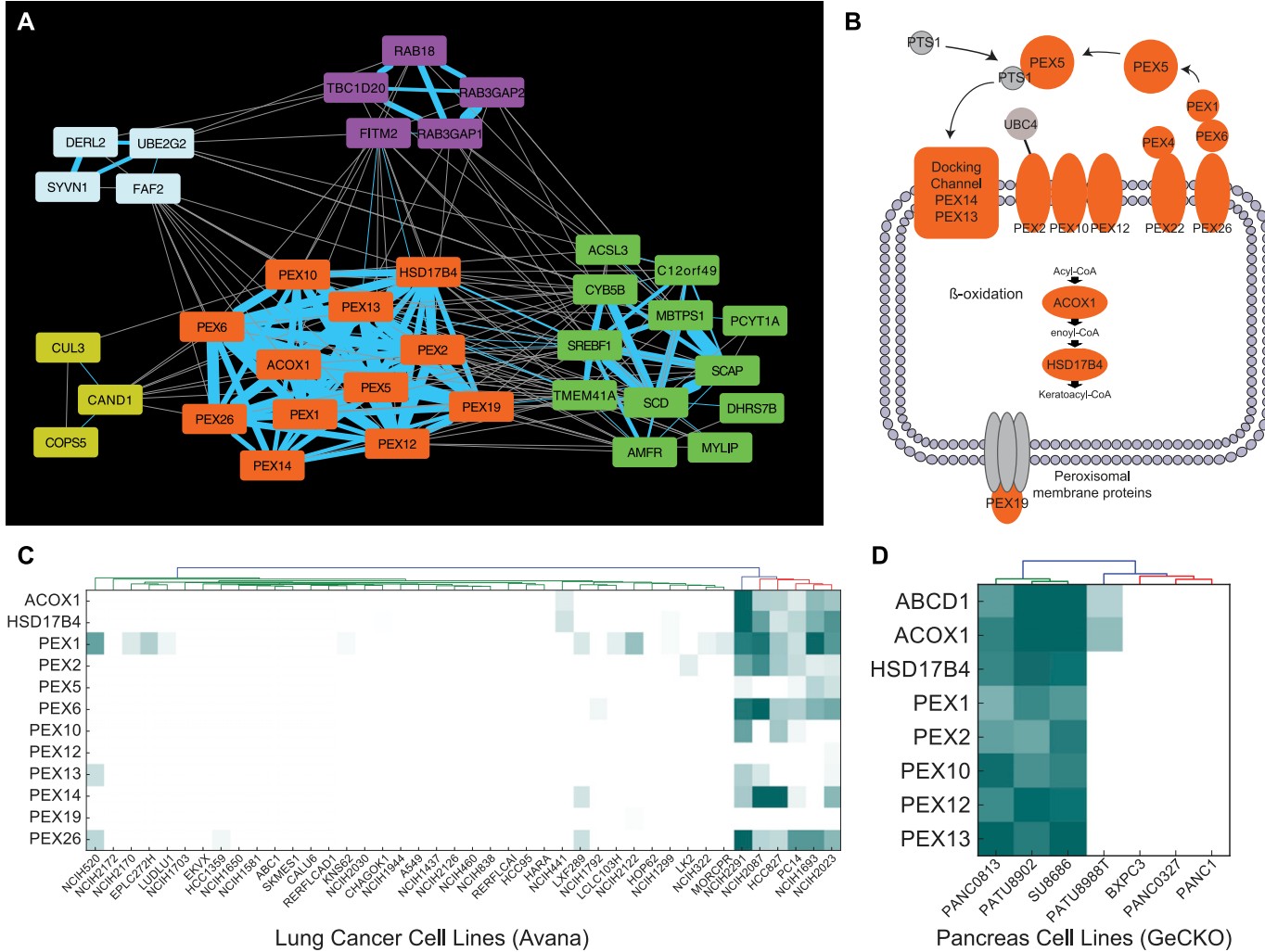

**Figure 5. Peroxisomal beta-oxidation.**
**(A)** A large group of peroxisome-associated genes (orange) is connected by high-correlation edges in the network (blue). This cluster is connected by less-stringent edges (Benjamini adjusted *P*-value < 0.01; gray edges) to other clusters containing sterol regulatory genes (green nodes) and the RAB18 GTPase (purple nodes). **(B)** The PEX cluster contains 12 genes, including two enzymes involved in fatty acid oxidation and 10 peroxisome biogenesis and maintenance genes. **(C, D)** The PEX cluster is emergently essential in a subset of lung cancer cell lines in the Avana data and (D) in a subset of pancreatic cancer cell lines in the GeCKO data.

of cellular amino acid levels (Fig 6A and B). Fig 6 shows the relationships between a series of network modules describing the core mTOR pathway and several regulatory modules. The mTOR cluster includes mTORC1/2 subunits *MTOR*, *MLST8*, *MAPKAP1*, and *RICTOR* (mTORC1-specific subunit *RAPTOR* is never essential and, therefore, absent from the network); canonical mTORC1/mTORC2 regulatory and signaling components *PDPK1*, *AKT1*, and *PIK3CB*; plus G-protein subunit *GNB2*, previously shown to physically interact with mTOR in response to serum stimulation (Robles-Molina et al, 2014). Canonical inhibition of mTOR by the *TSC1/TSC2* heterodimer—the *TSC1–TSC2* link is the top-ranked correlation in the entire data set, with $\rho$ = 0.93 ($P < 10^{-117}$)—is reflected in the anti-correlation of fitness profiles connecting the TSC1/2 cluster and the mTOR cluster.

*MTOR* response to cellular amino acid levels is modulated by an alternative pathway that functions at the lysosomal membrane (Bar-Peled & Sabatini, 2014). We identify a large cluster containing several genes involved in lysosomal protein and transport, including the *HOPS* complex (Balderhaar & Ungermann, 2013; Jiang et al, 2014) and the *VPS26/29/35* retromer complex (Hierro et al, 2007; Seaman, 2012). This strongly connected cluster also contains the Rag GTPases RagA (*RRAGA*) and RagC (*RRAGC*) that transmit information on amino acid abundance to mTORC1 (Bar-Peled & Sabatini, 2014). The Rag GTPases are in turn activated by the Ragulator complex (Sancak et al, 2010; Bar-Peled et al, 2012) and folliculin (*FLCN*) (M et al, 2017), also members of the cluster. The GATOR-1 complex is a nonessential suppressor of essential Rag GTPase activity (Bar-Peled et al, 2013) and is, therefore, absent from our network, but essential suppression of GATOR-1 by GATOR-2 (Bar-Peled et al, 2013; Wei et al, 2014) is reflected by the strong linkage of the GATOR-2 complex to both the Ragulator and mTOR complexes.

Within the MTOR meta-cluster, we further identify a complex containing three regulators of protein phosphatase 2A (*LCMT1*, *TIPRL*, and *PTPA*), whose strong connectivity to the *TSC1/2* complex may suggest a regulatory role for PP2A in MTOR signaling. PP2A has

previously been posited to be an activator of *TSC1/2* upstream of *MTOR* (Vereshchagina et al, 2008); the coessentiality network suggests specific PP2A regulators that may mediate this regulation.

A third example of the process-level interactions in cells demonstrates the hierarchy of operations required for post-translational maturation of cell surface receptors. Several clusters in our network describe the ER-associated glycosylation pathways (Fig 7A and B), including synthesis of lipid-linked sugars via the dolichol–phosphate–mannose (DPM) pathway (Ashida et al, 2006; Maeda & Kinoshita, 2008) and extension via the mannosyl-transferase family. Glycan chains are transferred to asparagine residues of target proteins via the N-oligosaccharyltransferase (OST) complex. Nascent polypeptide chains are glycosylated as they are cotranslationally translocated into the ER, a process facilitated by signal sequence receptor dimer *SSR1/SSR2*, and ER-specific Hsp90 chaperone *HSP90B1* facilitates proper folding. The OST complex and its functional partners are represented in a single large complex (Fig 7A). Both DPM and OST are highly connected to the large complex encoding glycosylphosphatidylinositol (GPI) anchor synthesis; DPM is required for GPI anchor production (Watanabe et al, 1998; Kinoshita & Inoue, 2000) before transfer to target proteins (Fig 7B).

A variety of oncogenic drivers among the cell lines underlying this network give rise to background-specific dependencies, including a variety of mutated and/or amplified RTKs with specific, and mutually exclusive, essentiality profiles. Insulin-like growth factor receptor *IGF1R* is one such RTK, which appears in a cluster with receptor-specific downstream-signaling proteins insulin-receptor substrate 1 and 2 (*IRS1* and *IRS2*). *IGF1R* is a highly N-glycosylated RTK, and the *IGF1R* complex is tightly connected to the OST complex in our network. *EGFR*, also highly glycosylated (Kaszuba et al, 2015), appears in its own cluster with signaling adapter protein *SHC1* and is also linked to the OST complex (Fig 7A) despite being mutually exclusive with *IGF1R* (Fig S2B). Interestingly, *EGFR* is more strongly connected with a separate complex involved in glycosphingolipid biosynthesis (that is itself

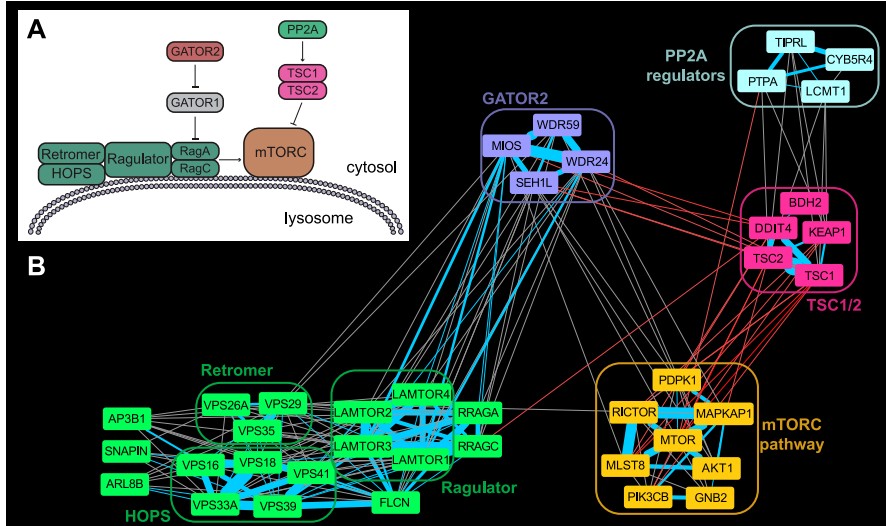

**Figure 6. mTORC pathway regulation.**
**(A)** The mTORC1/2 complexes are regulated by the canonical TSC1/2 pathway, but amino acid sensing is performed via the Ragulator complex at the lysosome. **(B)** Clusters in the coessentiality network represent components involved in mTORC regulation, and edges between clusters are consistent with information flow through the regulatory network (red edges indicate negative correlation).

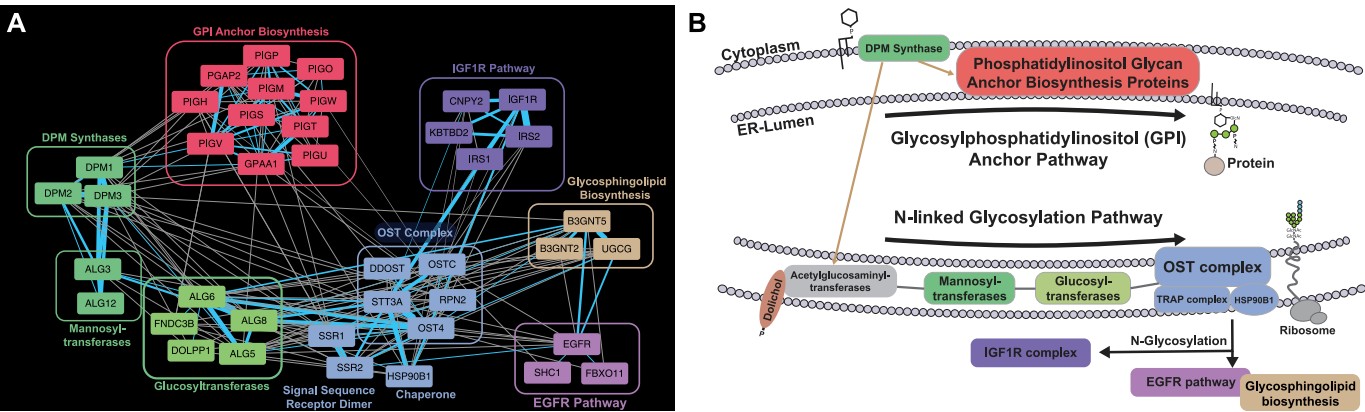

**Figure 7. Glycosylation of cell surface receptors.**
**(A)** Pathways involved in protein glycosylation and GPI anchor biosynthesis in the ER. **(B)** A network of clusters around glycosylation tracks the biogenesis and elongation of carbohydrate trees (DPM synthase, mannosyltransferases, and glucosyltransferases) to their transfer to target proteins via N-linked glycosylation by the OST complex. Cell surface glycoproteins EGFR and IGF1R are both strongly linked to the OST complex despite their essentiality being mutually exclusive in cell lines.

linked to the OST complex). Prior work suggests that membrane glycolipid composition can strongly influence *EGFR* autophosphorylation and signaling (Coskun et al, 2011). In contrast, fibroblast growth factor receptor *FGFR1* is absent from this meta-network but is strongly associated with heparin sulfate biosynthesis (Fig S2); HS is a known mediator of FGF receptor–ligand interaction (Wu et al, 2003).

# Conclusions

Systematic genetic interaction screens in yeast revealed that most genetic interactions occur either within a biological pathway or between related pathways. We demonstrate that single-gene fitness profiles across screens in genetically diverse human cell lines are analogous to genetic interaction screens across defined iso-genic query strains. Importantly, as with model organisms, human genes with correlated fitness profiles are highly likely to participate in the same biological process. We take advantage of this fundamental architectural feature of genetic networks to create a functional interaction map of bioprocesses that demonstrates information flow through a human cell. The network predicts gene function and provides a view of process-level interactions in human cells, allowing a level of abstraction beyond the gene-centric approach frequently used.

The network is derived from the emergent essentiality of defined biological processes and the genes required to execute them. We show that this approach significantly expands our knowledge beyond current networks of comparable design (e.g., STRING, HumanNet). Although the coessentiality network does not capture a large portion of protein–protein interactions (Chatr-aryamontri et al, 2017) or genetic interactions (Horlbeck et al, 2018), it predicts PPI with sensitivity comparable to coexpression networks (Fig S6).

Reconstructing biological processes from coessentiality information has some limitations. First of all, interactions between genes that do not affect cell fitness cannot be captured. Second, interactions between gene pairs that are not perturbed in the cell line pool cannot be captured. Last, the presence of genetic alterations like mutation or copy number amplification can generate confounding effects. A critical next step will be to understand the underlying context that drives the emergent essentiality of specific bioprocesses in specific backgrounds. The health implications of this question are profound. In cancer, to understand the causal basis of modular emergent essentiality is to identify matched pairs of biomarkers (the causal basis) and precision targets (the essential pathway) for personalized chemotherapeutic treatment. In addition, lineage-specific essential processes could provide explanatory power for germline mutations causing tissue-specific disease presentation, in cancer as well as other diseases.

Expanding the coverage of the network will require different screening approaches. Fitness screens in cancer cell lines in rich media will miss cellular dependencies that are present only under stress conditions. In yeast (Hillenmeyer et al, 2008) and nematodes (Ramani et al, 2012), these context-dependent fitness effects comprise the most of the genes in the genome. Increasing the coverage of the genetic interaction network beyond the ~3,000 genes whose fitness profiles covary across human cancer cell lines will require screening in different nutrients and perturbagens, as well as sampling genetic backgrounds outside common cancer genotypes. Nevertheless, the indirect approach to identifying genetic interactions from monogenic perturbation studies is demonstrably effective and offers a powerful tool for navigating the network of connections between cellular bioprocesses. The coessentiality network used in this study can be viewed interactively at https://hartlab.shinyapps.io/pickles/ (Lenoir et al, 2018) and downloaded at the NDEx project.

# Materials and Methods

### Construction of coessentiality network

A raw read count file of CRISPR pooled library screens for 342 cell lines using Avana library (Meyers et al, 2017, Avana project) was

downloaded from the data depository (https://figshare.com/articles/_/5520160). We only kept protein-coding genes for further analysis and updated their names using HGNC (Yates et al, 2017) and CCDS (Farrell et al, 2014) database. We discarded sgRNAs targeting multiple genes to prevent covariation from same sgRNA depletion. Raw read counts of each cell lines were analyzed through updated BAGEL v2 build 109 (https://github.com/hart-lab/bagel). In comparison with published BAGEL version v0.92 (Hart & Moffat, 2016), this updated version used a linear regression model at the step of calculating Bayes factor to overcome the dynamic range issue in the previous version (Fig S7).

First, we split cell lines by their experiment batch number with plasmid DNA control cell line of each batch. Then, normalized fold change was calculated for each batch as described by Hart and Moffat (2016), using default parameters (pseudo-count = 5 and minimum reads = 0). Essentiality of genes was calculated using gold standard reference sets of 684 core essential genes and 927 nonessential genes (Hart et al, 2014) (Hart et al, 2017). Lists of core essential genes and nonessential genes used in this study have been uploaded on the same repository with BAGEL v2 software. Among 341 cell lines (excluding a control cell line), three cell lines, ASPC1_PANCREAS, HEC59_ENDOMETRIUM, and U178_CENTRAL_NERVOUS_SYSTEM, failed to generate essentiality scores because fold changes of reference core essential genes and nonessential genes were indistinguishable. Last, we assembled essentiality profiles of 338 cell lines into a matrix.

We checked screen quality using "precision-recall" function in BAGEL software, and we calculated F-measure (BF = 5), which is the harmonic mean of precision and recall at Bayes factor 5. Then, among 338 cell lines, 276 cell lines were selected for further study, by F-measure (>0.85) and the number of essential genes (<2,000), to prevent noise from marginal quality of screens (Tables S2 and S3 and Fig S1). Essentiality of genes was preprocessed using quantile normalization within each cell line (Table S4). Then, correlation of essentiality of two genes was calculated using Pearson correlation coefficient (PCC) for all possible pairs. To remove false positives from variation of nonessential genes and copy number artifacts, we discarded genes essential in less than three cell lines among 276 cell lines and pairs of two genes located within 20M window on the same chromosome from the network. Finally, high significant relationships were selected by taking positive correlations with Bonferroni-corrected *P*-value less than 0.05. The strict-threshold coessentiality network is composed of 3,483 genes and 68,813 edges. In addition, we investigate interactions biased to off-target effects. We have tested all pairs whether the correlation of two genes drops after removing sgRNAs target the other gene allowing 1-bp mismatch. We marked all interactions that dropped below the threshold. We also constructed an extended coessentiality network including both positive and negative interactions cutoff by Benjamini–Hochberg adjusted *P*-value 0.01 (n = 285k positive and 149k negative correlations) (Table S5). All network figures shown in this article were drawn using Cytoscape (Shannon et al, 2003).

## Performance benchmark of coessentiality networks

Performance benchmark of coessentiality network was conducted by a pathway enrichment test using the KEGG pathway database (downloaded in 2015). Pathways having >200 annotated genes (including the ribosome, spliceosome, and proteasome) were discarded to minimize bias. For each network, gene pairs were ranked by PCC and binned into groups of 1,000 pairs. The cumulative LLS (Lee et al, 2011) was calculated per each bin as follows:

$$LLS = \frac{\text{The odds of within pathway interations in sample}}{\text{The odds of within pathway interactions in total possible pairs}}.$$

To compare performance with other screens, we constructed coessentiality networks for two other CRISPR data sets (Wang et al, 2015, 2017; Aguirre et al, 2016) and three shRNA data sets (Tsherniak et al, 2017; Hart et al, 2017b *Preprint*; McDonald et al, 2017). We downloaded a coessentiality network directly from Hart et al. For Aguirre et al (2016) and Tsherniak et al (2017) screens, we downloaded raw read count files from the article and Project Achilles site (Achilles CRISPR screens v3.3.8 and shRNA screens v2.19.1, https://portals.broadinstitute.org/achilles). For Wang et al screens, we downloaded raw read counts from their article. For McDonald et al (2017), we downloaded processed fold change files for three different pools from the DepMap database (https://depmap.org/portal/). We calculated Bayes factor profiles from raw read counts of screens and controls defined in articles through the BAGEL pipeline. To filter low-quality screens out, we applied thresholds of F-measure (BF = 5) 0.80 for Aguirre et al (2016) and Wang et al and F-measure (BF = 0) 0.60 and 0.70 for Tsherniak et al (2017) and McDonald et al (2017), respectively. We combined Bayes factor profiles from different pools of McDonald et al (2017) by taking average Bayes factor of overlapped genes. Then, we constructed coessentiality networks from Bayes factor profiles. We discarded pairs of two genes within the 20M window for networks from CRISPR screens to minimize the possibility of copy number artifacts.

## Detecting functional modules in coessentiality network

Network clustering was conducted with the Markov Cluster Algorithm (MCL) (Enright et al, 2002). We first converted a network file into a tab file and an mci file using "mcxload." Then, we ran MCL using various i-parameters. Last, we made a list of modules using "mcxdump." To determine the best i-parameter, we tested functional enrichment by measuring LLS of in-cluster pairwise connection against Gene Ontology Biological Process terms. The coessentiality network contains two dense and large clusters, which are mitochondrial oxidative pathway and mitochondrial ribosome subunits. To avoid bias from these two clusters, we excluded large clusters in test set (more than 50 genes) and large terms in true positive set (terms with more than 200 genes, and proteasome-, ribosome-, and spliceosome-related terms). Because we found that there is little difference between different parameters (data not shown), we decided to use the default parameter (I = 2.0). As a result, total 527 clusters were identified, 309 of them with at least three genes. Clusters are listed in supplementary data (Table S6). Pathway annotations of each cluster are summarized for Gene Ontology, KEGG, NCI_Nature, and Reactome in supplementary data (Table S7).

## Investigation of factors driving essentiality of clusters

To identify molecular genetic factors associated with cluster essentiality, we downloaded RNA -seq , copy number variation, and mutation profiles from the Cancer Cell Line Encyclopedia (CCLE) database (Barretina et al, 2012) in 2017. We classified cell lines as being "case" cells if the median BF of all genes in the cluster was >5, and all others as "control" cell lines. We then calculated the P-values of differential expression, copy number, and mutation between mean essentiality of clusters and gene properties. For RNA-seq expression, we used $\log_2(FPKM + 0.5)$. For copy number variation, we discretized copy number value into three classes (logCN ≥ 0.4 => amplified, logCN ≤ –0.4 => deleted, and –0.4 < logCN < 0.4 => neutral). For mutation data, we made a binary profile of mutation presence, treating silent mutations as wildtype. To measure significance of associated factors, we conducted a t test for expression data, Fisher's exact test for copy number (with amplifications and deletions calculated separately), and Fisher's exact test for binary mutation data.

## Identifying tissue-specific essentiality of clusters and genes

Tissue-specific essentiality was calculated for 517 clusters and 3,483 essential genes against 19 tissues: upper aerodigestive tract, thyroid, large intestine, autonomic ganglia, soft tissue, central nervous system, haematopoietic and lymphoid tissue, stomach, endometrium, liver, urinary tract, bone, lung, breast, skin, oesophagus, ovary, kidney, and pancreas. For each cluster, we first calculated mean essentiality of member genes per cell line. Then, for each tissue, we conducted Wilcoxon rank sum tests of two groups, a group belonging to the target tissue type, and a group consisting of all other tissue types. Last, adjusted P-value was measured by Bonferroni correction of P-value.

## Investigation of link overlap between other networks

To investigate how many interactions are overlapped with the coessentiality network from Avana screens, we downloaded functional interaction networks (HumanNet [Lee et al, 2011] and STRING v10 [Szklarczyk et al, 2015]) and protein–protein interaction networks (BioGRID [Chatr-aryamontri et al, 2017], CORUM complex [Ruepp et al, 2010]). For STRING v10, we used interaction threshold score 0.500. For CORUM complex data, we generated pair-wised interactions between protein members in the same protein complex. For fair comparison, genes not in the coessentiality network were excluded from the investigation.

## Protein complex comparison with a coexpression network from the same 276 cell lines

To construct a coexpression network, we downloaded RNA-seq expression profile from the CCLE database. Only matched cell lines and genes used for constructing the coessentiality network (276 cell lines) were kept for further steps. To prevent abnormal outlier or zero values, we gave 0.5 pseudo count to all genes, divided each value by the mean of each gene, and took the log form of

the resulting value. Expression similarity for all possible combinations of two genes was measured by PCC (Lee et al, 2011). Protein complex information was downloaded from CORUM database. We directly compared PCC values of interactions within a protein complex to observe differences between coessentiality network and coexpression network. Protein complexes having at least four interactions are only considered for comparison. We manually curated protein complexes with significant difference of average PCC (dPCC > 0.3) between coessentiality interactions and coexpression interactions, and then collapsed similar complexes into one representative. The pairs discarded in the filtering step of coessentiality network construction were not used in this comparison.

## Drug correlation and data curation

Drug log(IC50) values used for correlation analyses were taken from the Genomics of Drug Sensitivity in Cancer (GDSC) database (Yang et al, 2013). Data taken from TableS4A.xlsx located at: https://www.cancerrxgene.org/gdsc1000/GDSC1000_WebResources/Home.html.

Cell line annotation style from TableS4A was altered to match Avana project cell line annotation style. GDSC data contained log (IC50) values from 990 cell lines, which overlapped with 192/276 cell lines used in the Avana project. Log IC50 values contained 265 unique GDSC drug IDs with 250 unique drug names. Pearson correlations were computed using cor.test from the R package stats (version 3.2.3), based on mean gene BF in a cluster in a cell line against the matching cell line log IC50 value of each drug. Correlations calculated had between 34 and 187 data point pairs (mean BF, log IC50) within the overlapping 192 cell lines between GDSC database and Avana project. Pearson correlations that resulted in a negative correlation with a P-value less than $10^{-4}$ were added to the annotation text file. Negative correlations imply that IC50 values decrease as mean cluster BF in a given cell line increases (i.e., high BF implies increased sensitivity to drug). Correlations were calculated for the top 309 co-correlation ranked clusters.

## Investigation of cancer-specific genetic backgrounds

### Data preparation for annotation

The CCLE Reverse Phase Protein Array (RPPA) data, RPPA antibody information, and cell line annotations of 1,037 cancer cell lines were retrieved from the CCLE portal at: https://portals.broadinstitute.org/ccle/data. We used the gene-centric RMA-normalized expression data. Also, we used preprocessed RNA-seq and drug information (log IC50 data) used for other analysis in this article.

### MDM p53 cluster

Essentiality scores for each gene were preprocessed using quantile normalization within each cell line. The quantile-normalized essentiality scores for the selected genes for each of the 276 cell lines were gathered in a matrix. Cell lines with essentiality scores lower than or equal to –10 were set at a Bayes factor of –10.

Next, the heat map was plotted sorting the cell lines by the mean Bayes factors for each gene in the cluster by using the matplotlib package in Python. The heat map was annotated by the presence of

TP53 mutations (orange) with missing values as black and the Nutlin 3a natural log half maximal inhibitory concentration (IC50) values for each cell line with missing values as light grey.

### Neuroblastoma with MYC
From the 64k network, the *MYC* gene was added to Cluster 43 (MYCN) genes, and their quantile-normalized essentiality scores were gathered in a matrix for all high-quality Avana project cell lines. The heat map was plotted sorting the cell lines by the mean Bayes factors for each gene in the cluster. The heat map was annotated with MYC and MYCN expression values as well as a tissue key, specifying the neuroblastoma cell lines in orange.

### MYB-AML
From the 64k network, the quantile-normalized essentiality scores of genes from Cluster 17 were gathered in a matrix for all high-quality Avana project screens. The heat map was plotted sorting the cell lines by the mean Bayes factors for each gene in the cluster and annotated by a tissue key specifying the cell lines from the hematopoietic and lymphoid tissues in orange.

### BRCA subtypes
Clusters 52 and 55 from the 64k network were combined and the quantile-normalized essentiality scores for each gene in the clusters were gathered in a matrix after filtering for the cell lines from the breast tissue. The heat map was plotted by sorting with the mean essentiality scores for the genes in the clusters across the breast cell lines. The heat map was annotated with the log2 copy number values for the *ERBB2* gene, the RPPA values for ERBB2, expression values for ESR1 and CP724714, and Refametinib log IC50 values for each cell line with missing values as light grey.

### BRAF cluster
From the 32k network, quantile-normalized essentiality scores for the genes in cluster 19 were gathered in a matrix for all high-quality Avana cell lines. The cell lines were sorted by the mean Bayes factors and a heat map was plotted. The heat map was annotated with log2 copy number, RPPA values, presence of mutation (in orange) for *BRAF*, and the log IC50 values for PLX-4720, with missing values in light grey.

### RTK cluster
We filtered for the cell lines in which at least one of the following RTKs: EGFR, ERBB2, ERBB3, FGFR1, IRS2, and IGF1R have a BF of greater than 20. Then for that subset of cell lines, the essentiality scores of the RTKs and their downstream effectors were gathered in a matrix. Next, in Python, the hierarchical clustering package called scipy.cluster.hierarchy was used to cluster the cell lines for each RTK. We used the "average" method representing the UPGMA algorithm and the "Euclidean" distance metric to calculate the distance between the newly formed cluster and each remaining cluster and perform hierarchical clustering. The clustered heat maps were annotated for EGFR and ERBB2 copy number data.

### RAS cluster
We filtered for the cell lines in which at least one of the following genes: KRAS, NRAS, BRAF, and PIK3CA had a BF of greater than 20.

Next, we obtained the essentiality scores of the RTKs and their downstream effectors for the filtered cell lines. Later, we performed hierarchical clustering to cluster the cell lines for each RTK as explained above. Next, the clustered heat maps were annotated with mutation status for KRAS, NRAS, BRAF, and PIK3CA in orange.

### F measure versus cluster essentiality correlations

For each cluster in the network, the mean Bayes factor of the genes in that cluster was calculated to get a mean essentiality score for the cluster for each cell line. Then, the Pearson correlations and corresponding *P*-values were calculated using the scipy.stats.pearsonr from the scipy package based on the mean cluster essentiality score in a cell line against F-measure value of the matching cell line.

## Supplementary Information

## Acknowledgements

The authors would like to acknowledge the scientists, administrators, and funding agents behind the Cancer Dependency Map project. Without their commitment to rapid release of open access data, none of these works would have been possible. Funding was supplied by the NIH/NCI Cancer Center Support Grant P30 CA016672 (the Bioinformatics Shared Resource) and the Cancer Prevention Research Institute of Texas grant RR160032. T Hart is supported by NIH/NIGMS grant R35GM130119. E Kim is supported by an award from the Prostate Cancer Foundation.

### Author contributions

E Kim: conceptualization, data curation, software, formal analysis, visualization, and writing—review and editing.
M Dede: data curation, formal analysis, and visualization.
WF Lenoir: data curation, formal analysis, and visualization.
G Wang: data curation, formal analysis, and visualization.
S Srinivasan: data curation and formal analysis.
M Colic: data curation and formal analysis.
T Hart: conceptualization, supervision, funding acquisition, and writing—original draft, review, and editing.

### Conflict of Interest Statement

The authors declare that they have no conflict of interest.

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
