## [Reviewer comments · Life Science Alliance]

Life Science Alliance

A network of functional gene interactions in human cells from knockout fitness screens in cancer

Eiru Kim, Merve Dede, Walter Lenoir, Gang Wang, Sanjana Srinivasan, Medina Colic, and Traver Hart
DOI: <https://doi.org/10.26508/lsa.201800278>

Corresponding author(s): Traver Hart, University of Texas MD Anderson Cancer Center

Review Timeline:

Submission Date:	2018-12-12
Editorial Decision:	2018-12-14
Revision Received:	2019-02-19
Editorial Decision:	2019-02-21
Revision Received:	2019-03-15
Editorial Decision:	2019-03-29
Revision Received:	2019-04-03
Accepted:	2019-04-04

Scientific Editor: Andrea Leibfried

Transaction Report:

Please note that the manuscript was previously reviewed at another journal and the reports were taken into account in inviting a revision for publication at *Life Science Alliance*.

Reviewer #1 Review

This submission by Hart and colleagues analyzes several genome-wide screening data sets of cell viability, using both RNAi and CRISPR technologies, to detect genetic interactions. In brief, they are looking for sets of genes that correlate in their fitness effects across cell lines, as evidence of shared function. These correlations are then used to build a network of genetic interactions, which they show enriches for known interactions detected by orthogonal means (i.e. protein-protein interactions), providing confidence that novel interactions detected by this approach are likely to validate.

I have some suggestions for the authors to consider:

- 1) Could the authors comment on, and attempt to quantitate, the false negative rate of this approach? There are numerous well-known genetic interactions and protein complexes that are not detected. How should readers interpret their absence?
- 2) There are several pre-prints regarding additional corrections that can / should be made to these data to avoid incorrect conclusions. One, by Martin and colleagues, describes the potentially confounding effect of guides with multiple targets, and is cited by this manuscript (although on Page 7 line 176 - I believe SOX10 is the true hit and SOX9 is the false positive). Another pre-print, by Goncalves et al, examines the distinction between ploidy and copy-number in regards to the "cutting effect." This potential additional correction should at least be mentioned.
- 3) From Figure 1B, it appears that the McDonald data produced essentially no usable information for this approach. How is this possible? Related, what is the dotted line in Fig 1B, some measure of significance?
- 4) Line 134: "showed the strongest enrichment for co-functional gene pairs" - please give more detail on how these gene pairs were defined in the first place, and how the choice of previously-annotated gene sets may affect the result (i.e. StringDB vs. CORUM, etc.). The authors get to this point later, but it might be worth more text on this topic earlier in the manuscript, if possible.
- 5) On the website, it would be nice for a message to appear if the user searches for a gene that is not in any of the co-essential sets; as-is, it seems like the site is simply non-responsive.

Reviewer #2 Review

Report for Author:

In this manuscript the authors re-analyze genome-wide CRISPR screening data produced at the Broad Institute. Briefly, they cluster the gene essentiality matrix to identify co-essential gene groups, and go on to interpret the individual clusters, as well as links between them. Similar analyses and methods have been previously utilised in model organism literature, and more recently by Rauscher et al. Mol Sys Biol 2018.

Overall, the paper is a demonstration that these analyses can be done. As promised in the abstract, the main deliverable is a network. Several derived clusters are described in depth, with many known features recovered, as well as novel hypotheses suggested. Many similar analyses were (likely concurrently undertaken, but earlier) presented by Rauscher on a more limited dataset. The main claims pertain to properties of derived clusters; it is difficult for this reviewer to distinguish claims about biology that are not confounded by the particularities of the clustering approach used.

Major comments

-The methods are not completely clear:

-The improved BAGEL pipeline is unpublished and untested. How do the implemented changes affect performance?

-The datasets were quantile normalized. Given the range of up to +100, it cannot be against a Gaussian distribution; Methods was not helpful here, and I could not identify table S4. What was done, and what is the justification, given that BAGEL already has internal normalization against true positive and negative gene sets, and that correlation calculations are not affected by affine transformations?

-Tissue-specificity of clusters is ascertained in a circular way - first, the signal in the essentiality matrix structure (which includes tissue information) is used to form the clusters, and the tests then again make use of the same structure. Due to this double dipping, the p-values reported are not expected to be meaningful. This circularity makes interpretation difficult overall.

-All clustering algorithms, including MCL, have free parameters that determine the granularity of the resulting clustering. It thus could be that a slight relaxation of the parameter collapses the observed connections between clusters into single clusters, and certainly connected clusters will emerge if more granular clustering is enforced. In this light, is it even possible to reject the "network of biological processes" hypothesis using these data and approach?

Minor comments

- The language could be improved in places; phrases like "vastly more", "quantum leap", and "hugely increased" are imprecise enough to raise eyebrows in the introduction, but offend more in the results ("highly modular network", "strong functional coherence", "significantly more functional information", "discrete pattern of essentiality"), and puzzle in discussion ("demonstrate[d] information flow", "emergent essentiality").

- Figure 3E is missing or mislabeled

- Lines 299-300 invoke epistatic interactions - it is not clear what alleles are proposed to be interacting.

Reviewer #3 Review

In this manuscript Kim et al. create a cancer coessentiality network from gene essentiality screens in cancer cell lines. They use the Achilles Avana dataset after showing that it has higher functional information content than multiple other shRNA and CRISPR/Cas9 datasets. After defining modules in the network they demonstrate the functional coherence of numerous modules and the functional relationships between modules.

While this work is not conceptually novel, it further demonstrates the value of gene essentiality data to studies of gene function and biological processes.

As the authors note, correlations between the essentiality profiles of two genes can be driven by the CRISPR DNA cutting effect ("CN effect"). Using one of the existing methods for correction of this effect instead of limiting the coessentiality network to include only pairs of genes at least 20MB away from each other would make it more complete. In addition, using an empirical null distribution composed of correlations of random pairs of genes to derive p-values might help correct for this effect as well.

The authors used different metrics and thresholds to define and filter low-quality shRNA/sgRNA screens in the various datasets they considered. As the authors compare the datasets, they should explain the choices of the filtering criteria or use a consistent one for all datasets.

Minor comments:

The network visualization of Fig 1C is not very informative, especially as gene names are illegible. The authors should use the latest publicly available Avana dataset (I think the last one has close to 500 cell lines). In addition, the Broad recently made public (<https://doi.org/10.1101/305656>) a combined Achilles/DRIVE RNAi dataset ("DEMETER2") that would be interesting to add to the comparison, especially of Fig 1b.

There are inconsistencies in the report of the size of the coessentiality network between the main and supplementary texts.

In Fig 3A changing the y-axis labels to say "index" instead of "number" would make it easier to understand that these are not bar plots but rather heatmap-like plots. The legend of Fig 3BCD should also be clarified to indicate what the numbers in the plots represent.

December 14, 2018

Re: Life Science Alliance manuscript #LSA-2018-00278-T

Dr Traver Hart
University of Texas MD Anderson Cancer Center
Dept of Bioinformatics and Computational Biology
1400 Pressler Unit 1410
Houston, Texas 77030

Dear Dr. Hart,

Thank you for transferring your manuscript entitled "Hierarchical organization of the human cell from a cancer coessentiality network" to Life Science Alliance. The manuscript was assessed by expert reviewers at another journal before, and the editors transferred those reports to us with your permission.

The reviewers thought that your work demonstrates the value of gene essentiality data for studying gene function and biological processes and that it provides a useful resource to others, but judged the conceptual advance as not very high. The latter is not a concern for publication in Life Science Alliance, and we would thus like to invite you to provide a revised version of your work for publication here. Importantly, the reviewers provide constructive input on how to improve your work and on how to better correct the dataset/exclude confounding effects and report a false-negative rate by re-analysis as well as by clarifying some aspects. We would thus invite you to provide a point-by-point response to the concerns raised and accordingly changes to the manuscript text and data. Please get in touch in case you would like to discuss individual revision points further.

Thank you for this interesting contribution to Life Science Alliance. We are looking forward to receiving your revised manuscript.

Sincerely,

Andrea Leibfried, PhD
Executive Editor
Life Science Alliance

Meyerhofstr. 1
69117 Heidelberg, Germany
t +49 6221 8891 502
e a.leibfried@life-science-alliance.org
www.life-science-alliance.org

- A letter addressing the reviewers' comments point by point.
- An editable version of the final text (.DOC or .DOCX) is needed for copyediting (no PDFs).
- High-resolution figure, supplementary figure and video files uploaded as individual files: See our detailed guidelines for preparing your production-ready images, <http://life-science-alliance.org/authorguide>
- Summary blurb (enter in submission system): A short text summarizing in a single sentence the study (max. 200 characters including spaces). This text is used in conjunction with the titles of papers, hence should be informative and complementary to the title and running title. It should describe the context and significance of the findings for a general readership; it should be written in the present tense and refer to the work in the third person. Author names should not be mentioned.

B. MANUSCRIPT ORGANIZATION AND FORMATTING:

Full guidelines are available on our Instructions for Authors page, <http://life-science-alliance.org/authorguide>

Reviewer #1:

This submission by Hart and colleagues analyzes several genome-wide screening data sets of cell viability, using both RNAi and CRISPR technologies, to detect genetic interactions. In brief, they are looking for sets of genes that correlate in their fitness effects across cell lines, as evidence of shared function. These correlations are then used to build a network of genetic interactions, which they show enriches for known interactions detected by orthogonal means (i.e. protein-protein interactions), providing confidence that novel interactions detected by this approach are likely to validate.

We thank the reviewer for these positive comments.

I have some suggestions for the authors to consider:

1) Could the authors comment on, and attempt to quantitate, the false negative rate of this approach? There are numerous well-known genetic interactions and protein complexes that are not detected. How should readers interpret their absence?

We appreciate the reviewer's question. We've checked false negative rate (1-TPR) of our coessentiality network (Supplementary Figure 6). Regarding results, we could found that only 4~8% of Protein interactions and 3~6% of genetic interactions are recapitulated by our coessential network. First of all, since coessentiality depends on fitness variability across cancer cell lines, our coessentiality network is limited to reconstructing interactions among genes/proteins that are essential in cancer cells. Secondly, genetic alterations like mutation and copy number alteration can generate confounding effect. Measuring coexpression is an alternative method of predicting protein-protein interactions. Our coessentiality network captures protein-protein or genetic interactions with sensitivity similar to a coexpression network with the same number of edges, constructed using RNA-seq data from same pool of cells. Furthermore, the integrated network of coessentiality and coexpression networks shows lower false negative rate. This result also confirms that coexpression and coessentiality contain complementary information.

We've added this description in the revised manuscript as follows.

“Although, coessentiality network couldn't capture a large portion of protein-protein or genetic interactions, it is comparable with coexpression networks that are often regarded as an alternative way of identifying protein-protein interactions. Reconstructing biological processes from coessentiality information has some limitations. First of all, interactions between genes that do not affect cell fitness aren't able to be captured. Secondly, interactions between two genes that aren't perturbed in the cell line pool aren't able to be captured. Lastly, the presence of genetic alterations like mutation or copy number amplification have the possibility to generate confounding effects. ”

2) There are several pre-prints regarding additional corrections that can / should be made to these data to avoid incorrect conclusions. One, by Martin and colleagues, describes the potentially confounding effect of guides with multiple targets, and is cited by this manuscript (although on Page 7 line 176 - I believe SOX10 is the true hit and SOX9 is the false positive). Another pre-print, by Goncalves et al, examines the distinction between ploidy and copy-number in regards to the "cutting effect." This potential additional correction should at least be mentioned.

We appreciated reviewer's constructive suggestion. We've analyzed potential off-target effects by tracking off-targets of sgRNAs and measured correlation drop after removing multitarget sgRNAs. Suspected off-target interactions are marked in Supplementary Table 6 (interaction data). Also, we've added this information in Figure 2B.

3) From Figure 1B, it appears that the McDonald data produced essentially no usable information for this approach. How is this possible? Related, what is the dotted line in Fig 1B, some measure of significance?

We apologize for incorrect analysis. We've reanalyzed the McDonald et al. network and corrected Figure 1B. Also we've added short description for the dotted line. "Cut-off of Meyers et al."

4) Line 134: "showed the strongest enrichment for co-functional gene pairs" - please give more detail on how these gene pairs were defined in the first place, and how the choice of previously-annotated gene sets may affect the result (i.e. StringDB vs. CORUM, etc.). The authors get to this point later, but it might be worth more text on this topic earlier in the manuscript, if possible.

We apologize for the insufficient description. We've added additional description

"We employed log-likelihood score to described the significance of enrichment. Each score was measured by 1,000 pairs cumulatively against background probability of true positive set, KEGG."

5) On the website, it would be nice for a message to appear if the user searches for a gene that is not in any of the co-essential sets; as-is, it seems like the site is simply non-responsive.

We've updated the website to show a message with gene name.

Reviewer #2:

In this manuscript the authors re-analyze genome-wide CRISPR screening data produced at the Broad Institute. Briefly, they cluster the gene essentiality matrix to identify co-essential gene groups, and go on to interpret the individual clusters, as well as links between them. Similar analyses and methods have been previously utilised in model organism literature, and more recently by Rauscher et al. Mol Sys Biol 2018.

Overall, the paper is a demonstration that these analyses can be done. As promised in the abstract, the main deliverable is a network. Several derived clusters are described in depth, with many known features recovered, as well as novel hypotheses suggested. Many similar analyses were (likely concurrently undertaken, but earlier) presented by Rauscher on a more limited dataset. The main claims pertain to properties of derived clusters; it is difficult for this reviewer to distinguish claims about biology that are not confounded by the particularities of the clustering approach used.

We appreciate reviewer's comments. In fact, Rauscher et al. is prediction of genetic interaction using genetic perturbation. Our coessentiality network is based on similar concept with a coexpression network to reconstruct functional pathway, and we categorized our coessential network as functional network. We've checked how many interactions are overlapped with Rauscher et al. and Pan et al., which examines coessentiality for protein complexes (Supplementary Figure 6). It shows that our coessentiality network contains different context with Rauscher et al., while our coessential network shares significant interactions with Pan et al.

To avoid confusion, we've added description in the manuscript as follows.

“Also, we checked overlap with previously published protein complex scale coessentiality network and inferred genetic interaction network using CRISPR screen data (Pan et al.;

Rauscher et al., 2018). We found there is a very few overlap between coessentiality networks and Rauscher et al. This discrepancy explains different scope of networks.”

Major comments

- The methods are not completely clear:
- The improved BAGEL pipeline is unpublished and untested. How do the implemented changes affect performance?

We apologize for lack of information. We’ve included comparison plots between V1 and V2 performance.

- The datasets were quantile normalized. Given the range of up to +100, it cannot be against a Gaussian distribution; Methods was not helpful here, and I could not identify table S4. What was done, and what is the justification, given that BAGEL already has internal normalization against true positive and negative gene sets, and that correlation calculations are not affected by affine transformations?

We understand the reviewer’s concern. BAGEL measures essentiality based on distribution of fold changes of gRNA targeting reference core-essential genes and non-essential genes. Using pre-defined reference set, BAGEL can derive precisely not only essentiality of gene but also significance of the essentiality. Since coessentiality is measured by co-perturbed cell fitness by two genes, the correlation must be influenced by quality of screen. Thus, we employed quantile normalization to mitigate the effect of quality difference. This method also used in Rauscher et al. for same reason. We’ve revised the manuscript as follows

“We removed nontargeting and nonhuman gene controls and quantile normalized each data set to mitigate screen quality bias, yielding an essentiality score where a positive value indicates a strong knockout fitness defect and a negative value generally implies no phenotype”

Quantile normalization is performed without a reference distribution, where the value for the “n”th ranked gene is the median of the “n”th ranked gene across all samples.

- Tissue-specificity of clusters is ascertained in a circular way - first, the signal in the essentiality matrix structure (which includes tissue information) is used to form the clusters, and the tests then again make use of the same structure. Due to this double dipping, the p-values reported are not expected to be meaningful. This circularity makes interpretation difficult overall.

We apologize for the ambiguous description about the tissue-specificity of clusters. We agree that tissue specific clusters should be derived when we use an essentiality profile. However, it is hard to see this as circular logic. No tissue information is used to learn either the network or the clusters; the KEGG pathways used are tissue agnostic. Moreover, tissue-specific clusters are typically depleted for functional enrichment; e.g. tissue-specific transcription factors (found in every tissue-specific cluster) are rarely, if ever, included in functional annotation tests.

We also note in the text:

“Given the underlying data, it is not surprising that oncogenic signatures are clearly evident in the coessentiality network. However, the vast majority of the network structure does not appear to be driven by tissue specificity or mutational signatures.” (p8)

- All clustering algorithms, including MCL, have free parameters that determine the granularity of the resulting clustering. It thus could be that a slight relaxation of the parameter collapses the observed connections between clusters into single clusters, and certainly connected clusters will emerge if more granular clustering is enforced. In this light, is it even possible to reject the "network of biological processes" hypothesis using these data and approach?

We appreciate reviewer’s valuable comment and apologize for the lack of explanation. We understand that granularity is determinant of the size of cluster and can affect clustering result. We tried many different granularity and tested how the clusters recapitulate real biological process (GOBP). Since large clusters including mitochondrial oxidative pathway and ribosome genes, we benchmarked MCL results limited to clusters comprised of equal or below than 50 genes. Regarding the benchmark data, we couldn’t find significant difference of functional enrichment between 1.4~3.0. Thus, we determined to keep default parameter.

We’ve added a description into Supplementary Methods.

“To determine best i-parameter, we tested functional enrichment by measuring log likelihood score of in-cluster pairwise connection against Gene Ontology Biological Process terms. The coessentiality network contains two dense and large clusters, which are mitochondrial oxidative pathway and mitochondrial ribosome subunits. To avoid bias from these two clusters, we excluded large clusters in test set (over 50 genes) and large terms in true positive set (terms with over 200 genes, and proteasome, ribosome, and spliceosome related terms). Since we found that there is little difference between different parameters (Data not shown), we decided to use the default parameter (I=2.0).”

Minor comments

- The language could be improved in places; phrases like "vastly more", "quantum leap", and "hugely increased" are imprecise enough to raise eyebrows in the introduction, but offend more in the results ("highly modular network", "strong functional coherence", "significantly more functional information", "discrete pattern of essentiality"), and puzzle in discussion ("demonstrate[d] information flow", "emergent essentiality").
- Figure 3E is missing or mislabeled

We've clarified the language in several locations.

- Lines 299-300 invoke epistatic interactions - it is not clear what alleles are proposed to be interacting.

We refer to epistatic relationships between genes, not specific alleles.

Reviewer #3:

In this manuscript Kim et al. create a cancer coessentiality network from gene essentiality screens in cancer cell lines. They use the Achilles Avana dataset after showing that it has higher functional information content than multiple other shRNA and CRISPR/Cas9 datasets. After defining modules in the network they demonstrate the functional coherence of numerous modules and the functional relationships between modules.

While this work is not conceptually novel, it further demonstrates the value of gene essentiality data to studies of gene function and biological processes.

We appreciate reviewer's careful review.

As the authors note, correlations between the essentiality profiles of two genes can be driven by the CRISPR DNA cutting effect ("CN effect"). Using one of the existing methods for correction of this effect instead of limiting the coessentiality network to include only pairs of genes at least 20MB away from each other would make it more complete. In addition, using an empirical null distribution composed of correlations of random pairs of genes to derive p-values might help correct for this effect as well.

We appreciate reviewer's comment. Recently, several papers have discussed copy-number effect correction on the fold-change level. These can be helpful to build unbiased network as an alternative way. We've checked that current method we employed in this manuscript also filtered copy number alteration out significantly. While our coessentiality network reconstructs ~4% of protein-protein interaction, the interactions mis-filtered by distance threshold contain only ~0.07% of possible known protein-protein interactions. It confirms that the distance filtering step doesn't lose much information.

The authors used different metrics and thresholds to define and filter low-quality shRNA/sgRNA

screens in the various datasets they considered. As the authors compare the datasets, they should explain the choices of the filtering criteria or use a consistent one for all datasets.

Usually APR and F-measure showed correlation. But, we changed strategy of cut-off from APR to F-measure to keep consistency. However, because of different characteristic between CRISPR and shRNA screen, we used less stringent thresholds for shRNA screens than the one used for CRISPR screens.

Minor comments:

The network visualization of Fig 1C is not very informative, especially as gene names are illegible.

Figures with this level of granularity are common in manuscripts featuring network-driven analyses. We cannot add 3,000 gene names to the figure, but we have manually added cluster annotation labels.

The authors should use the latest publicly available Avana dataset (I think the last one has close to 500 cell lines). In addition, the Broad recently made public (<https://doi.org/10.1101/305656>) a combined Achilles/DRIVE RNAi dataset ("DEMETER2") that would be interesting to add to the comparison, especially of Fig 1b.

We appreciate reviewer's comment. We've reanalyzed drive screens using data provided by depmap.org (Figure 1B).

There are inconsistencies in the report of the size of the coessentiality network between the main and supplementary texts.

We've corrected the manuscript.

In Fig 3A changing the y-axis labels to say "index" instead of "number" would make it easier to

understand that these are not bar plots but rather heatmap-like plots. The legend of Fig 3BCD should also be clarified to indicate what the numbers in the plots represent.

We appreciate helpful suggestion. We've corrected the figure.

February 21, 2019

RE: Life Science Alliance Manuscript #LSA-2018-00278-TR

Dr. Traver Hart
University of Texas MD Anderson Cancer Center
Dept of Bioinformatics and Computational Biology
1400 Pressler Unit 1410
Houston, Texas 77030

Dear Dr. Hart,

Thank you for submitting your revised manuscript entitled "A network of functional gene interactions in human cells from knockout fitness screens in cancer". As you will see, one of the original reviewers saw your manuscript again and appreciates the introduced changes. We appreciate them as well and would thus be happy to publish your paper in Life Science Alliance pending final revisions necessary to meet our formatting guidelines:

- please link your ORCID iD to your profile in our submission system
- please add callouts in the manuscript text to Fig4 panel E; Fig6 panels A and B; Fig7B; SFig2 panels A, C and D; and to SupplTables 2, 3, 4, 5
- please incorporate (for easier discoverability by readers) the methods into the main text; note that the references mentioned in the current supplement do not list 10 authors et al. The legends for the S figures can also get incorporated in the main manuscript docx file and the S figures should get uploaded as individual files.

A. FINAL FILES:

B. MANUSCRIPT ORGANIZATION AND FORMATTING:

Sincerely,

Reviewer #2 (Comments to the Authors (Required)):

The Authors have comprehensively addressed all of my previous concerns.

This is a nice study, that while descriptive, gives important insight into what can be achieved by aggregating data from many genome-wide knock-out screens. As thousands of such experiments are going to be executed over the coming years, this paper will serve as the landmark of the types of analyses that ought to be conducted.

March 29, 2019

RE: Life Science Alliance Manuscript #LSA-2018-00278-TRR

Dr. Traver Hart
University of Texas MD Anderson Cancer Center
Dept of Bioinformatics and Computational Biology
1400 Pressler Unit 1410
Houston, Texas 77030

Dear Dr. Hart,

Thank you for submitting your revised manuscript entitled "A network of functional gene interactions in human cells from knockout fitness screens in cancer". We would be happy to publish your paper in Life Science Alliance pending fixing the following issues:

- a table from the previous version of your manuscript is now missing (see my previous emails)
- author Gang Wang is missing from the author list in the submission system, please add again

A. FINAL FILES:

B. MANUSCRIPT ORGANIZATION AND FORMATTING:

Sincerely,

April 4, 2019

RE: Life Science Alliance Manuscript #LSA-2018-00278-TRRR

Dr. Traver Hart
University of Texas MD Anderson Cancer Center
Dept of Bioinformatics and Computational Biology
1400 Pressler Unit 1410
Houston, Texas 77030

Dear Dr. Hart,

Thank you for submitting your Research Article entitled "A network of functional gene interactions in human cells from knockout fitness screens in cancer". It is a pleasure to let you know that your manuscript is now accepted for publication in Life Science Alliance. Congratulations on this interesting work.

*****IMPORTANT:** If you will be unreachable at any time, please provide us with the email address of an alternate author. Failure to respond to routine queries may lead to unavoidable delays in publication.*******

DISTRIBUTION OF MATERIALS:

Again, congratulations on a very nice paper. I hope you found the review process to be constructive and are pleased with how the manuscript was handled editorially. We look forward to future exciting submissions from your lab.

Sincerely,
